# Hierarchical Gradient-Informed Reinforcement Learning for Scalable and Partially Observable Dynamic Resource Allocation

## Abstract

Dynamic resource allocation problems (DRAPs) are prevalent in critical domains like transportation and energy management, and can be naturally modeled as dynamic systems, posing challenges in scalability and partial observability. We propose a novel framework, **Hierarchical Gradient-Informed Reinforcement Learning (HGRL)**, which integrates hierarchical multi-agent reinforcement learning with a Global Demand Inference Network (GDI-Net). HGRL decomposes DRAPs into multi-scale subproblems, enabling scalable decision-making across large environments. GDI-Net addresses partial observability by inferring and identifying multi-scale global demand and directional gradients from local agent observations, enhancing policy awareness and guiding exploration. Experiments on synthetic and real-world datasets demonstrate that HGRL significantly outperforms strong baselines, achieving up to **55.1%** improvement in demand coverage and **35.5%** improvement in transportation efficiency on the real-world dataset. **Code is available at:** https://anonymous.4open.science/r/HGRL4DRA-B40FS/

## 1 Introduction

Efficient resource allocation is a fundamental problem across various domains, including transportation, logistics, energy management, and communication systems. In these settings, optimizing the distribution of resources, such as vehicles, energy units, or servers, ensures that limited resources meet dynamic demands while minimizing operational costs. For instance, in ride-hailing platforms, allocating vehicles to customers in real-time improves user satisfaction and fleet efficiency. As the scale and complexity of these systems continue to grow, solving Resource Allocation Problems (RAP) in real-time becomes increasingly important, especially under strict time constraints.

Depending on the temporal characteristics of the system, RAP can be divided into static and dynamic categories. While static resource allocation Gendron & Semet (2009); Chen et al. (2010); Jarboui et al. (2013) assumes fixed and known demands over time, many real-world applications exhibit high variability and uncertainty, making dynamic resource allocation problems (DRAPs) more representative of practical challenges. DRAPs are common in environments such as smart grids, transportation networks, and emergency response systems, where conditions change rapidly and decisions must be made in real-time to fulfill the demand.

Over the years, dynamic resource allocation problems (DRAPs) have been addressed using traditional methods like Mixed Integer Linear Programming (MILP) and Dynamic Programming. MILP offers exact solutions under predefined constraints and has been applied to lane assignment, bus scheduling, and rail operations He et al. (2018); Wu et al. (2021); He & Peeta (2014). Markov Decision Process (MDP) -based approaches enable sequential decision-making under uncertainty and have been used in traffic signal control and vehicle dispatching Liu et al. (2019); Ruan et al. (2020); Liu et al. (2022).

Recently, Deep Reinforcement Learning (DRL) has emerged as a flexible alternative for handling complex and uncertain environments Saadi et al. (2025). By learning from interaction, RL adapts to dynamic traffic patterns beyond rule-based strategies. Applications in public transit and bike sharing Wu et al. (2024); Liu et al. (2023); Guo et al. (2024); Pan et al. (2019b) show that RL can effectively adjust scheduling policies in response to fluctuating demand, outperforming fixed-frequency approaches.

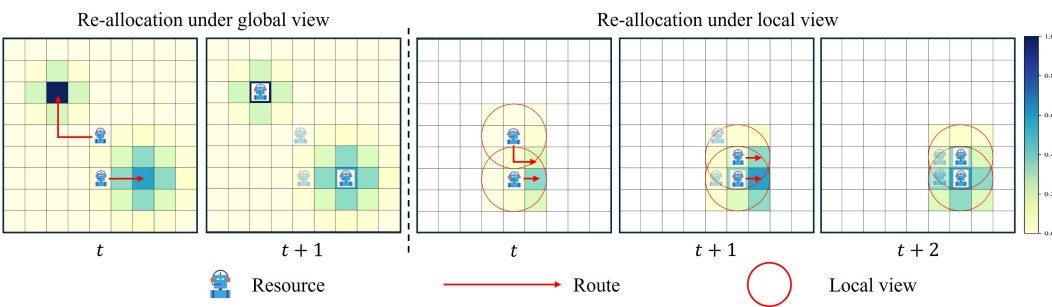

Figure 1: Illustration of dynamic resource allocation under global and local views

Despite their promise, solving dynamic resource allocation problems (DRAPs) at scale presents a major modeling challenge — **scalability**. In DRL-based approaches, the key to effective learning lies in capturing the joint dynamics of agents and resources. However, as the number of agents and resources increases, the size of the state and action spaces grows combinatorially, leading to exponential growth in complexity. This combinatorial explosion renders naive exploration strategies inefficient and makes training increasingly impractical for large-scale systems. The second challenge is **partial observability**, as real-world systems suffer from limited sensing, communication delays, or disruptions. As shown in Figure 1, agents with global views can directly target high-demand areas, while those with only local observations must iteratively adjust, resulting in slower and less effective responses. Although earlier methods perform well under ideal conditions, they often fail to make robust decisions when dealing with noisy or incomplete information—common in large-scale resource allocation scenarios.

To address the dual challenges of scalability and partial observability in dynamic resource alloca-tion problems (DRAPs), we propose a hierarchical multi-agent reinforcement learning framework integrated with a Global Demand Inference Network (GDI-Net). To tackle the **scalability** issue, our framework models DRAPs as a hierarchical multi-agent MDP, drawing inspiration from dynamic programming. Higher-level agents operate at coarse spatial resolutions to make long-term strategic decisions, while lower-level agents handle fine-grained execution. This hierarchical decomposition reduces the effective dimensionality of the state and action spaces, enabling efficient and scalable decision-making across large-scale systems. To mitigate **partial observability**, GDI-Net infers multi-scale global demand and gradient information from agents' local observations. By embedding this inferred information—computed via Sobel convolutional kernels—into the agents' state repre-sentations, the system enhances global awareness, guides exploration through spatial gradients, and improves overall policy performance under limited or noisy observations. The contributions of this paper are as follows:

- We propose a **Hierarchical Gradient-Informed Reinforcement Learning** (**HGRL**) under multi-agent reinforcement learning framework for dynamic resource allocation, modeling DRAP as a multi-level multi-agent MDP problem. This approach leverages dynamic programming to manage high-dimensional state and action spaces efficiently, enabling scalable decision-making.

- We introduce a novel state embedding method that utilizes the Global Demand Inference Network (GDI-Net) to infer multi-scale global demand and gradient information from local observations. This method allows agents to make more informed decisions even with limited local visibility.

- We validate our approach through experiments on real-world datasets, demonstrating the effective-ness of the proposed framework in improving training efficiency and decision-making accuracy for dynamic resource allocation problems.

## 2 RELATED WORK

Early DRAP solutions often relied on hand-crafted heuristics and classical optimization. Rule-based or greedy algorithms Sulistyo & Setiawan (2025); Datta & Pan (2021); Kumar et al. (2021) are simple and fast but struggle in highly dynamic settings due to fixed decision rules. More advanced formulations such as Mixed-integer linear programming (MILP) He et al. (2018); Wu et al. (2021); He & Peeta (2014) can capture rich constraints and yield optimal allocations for static snapshots, yet

solving them exactly is NP-hard and becomes intractable as problem size grows. These approaches lack the adaptability and scalability required for real-time, large-scale DRAPs.

To address these limitations, recent works have adopted RL, which has shown strong adaptability in domains such as power grids and transportation Ren et al. (2022); Wu et al. (2024; 2020); Wei et al. (2019); Liu et al. (2023); Guo et al. (2024). By learning policies from interaction, RL can adapt to changing demand and resource conditions without explicit modeling of all constraints. However, directly applying flat MARL to large-scale DRAPs suffers from two key challenges: (*i*) the joint state-action space grows exponentially with the number of agents, and (*ii*) coordination becomes difficult in partially observable settings without costly global communication.

To improve scalability and coordination, some works enhance MARL with stronger spatial representations. GridNet Han et al. (2019) employs a grid-wise encoder-decoder with stacked convolutions, embedding the global demand map and enabling implicit coordination via large receptive fields. Other approaches adopt explicit structured or hierarchical designs. For instance, HMF Yu (2023) uses a two-level hierarchy: attention-based mean-field Q-networks for intra-group coordination and a high-level Q-network for inter-group coordination. MDPO Ma et al. (2024) addresses partially observable, graph-structured environments through decentralized policies and hop-$k$ neighborhood communication, reducible to a fully connected graph in our case. While effective, these methods often rely on fixed embeddings or predefined graphs, which limit adaptability in highly dynamic and partially observable DRAPs. To our knowledge, we are the first to integrate GDI-Net's multi-scale demand embedding with RL in a joint training framework, enabling scalable coordination and robust decision-making under partial observability.

## 3 PROBLEM STATEMENT

In a discrete grid environment $G$, there exists a set of time periods $P = \{1, 2, \ldots, |P|\}$. For each period $p \in P$, each grid cell $g \in G$ is associated with a demand value $D_{p,g}$. Each resource $n \in N$ has an initial location $l_n^p \in G$, a fixed service capacity $U$, and a spatial service window of size $w \times w$. Let $l_n^p$ denote the position of resource $n$ at period $p$, and let $\mathcal{W}(l_n^p) \subseteq G$ denote the set of grid cells covered by the service window centered at $l_n^p$. The amount of demand resource $n$ can serve at grid cell $g$ in period $p$ is $\min\{D_{p,g}, U\}$.

Our objective is to determine the routes of resources $\{l_n^p\}_{n \in N, p \in P}$ that maximize the total covered demand while minimizing the cumulative travel distance:

$$\max_{\{l_n^p\}} \sum_{p \in P} \sum_{n \in N} \sum_{g \in \mathcal{W}(l_n^p)} \min\{D_{t,g}, U\} - \delta \sum_{p=2}^{|P|} \sum_{n \in N} \text{Dist}(l_n^{p-1}, l_n^p) \tag{1}$$

Here, $\text{Dist}(l_n^{p-1}, l_n^p)$ denotes the movement cost (e.g., Manhattan distance) for resource $n$ between consecutive periods, and $\delta > 0$ is a weight parameter balancing service utility and movement cost.

**Complexity.** The above DRAP problem is NP-hard. Proof is provided in Appendix A.

## 4 METHODOLOGY

Figure 2 illustrates the architecture of our proposed model, HGRL which comprises a Global Demand Inference Network and a multi-level Value-Decomposition Network. This design mitigates partial observability by inferring global demand from local observations Zhong et al. (2025) and estimating value functions across multiple abstraction levels. GDI-Net adopts the centralized training with decentralized execution (CTDE) paradigm, enabling agents to access global context during training while relying solely on local observations and inferred demand during execution.

### 4.1 HIERARCHICAL MULTI-AGENT MDP FRAMEWORK

Building on the hierarchical design principle introduced in FuNs Vezhnevets et al. (2017), which decomposes control across temporal scales, we formulate a hierarchical multi-agent reinforcement

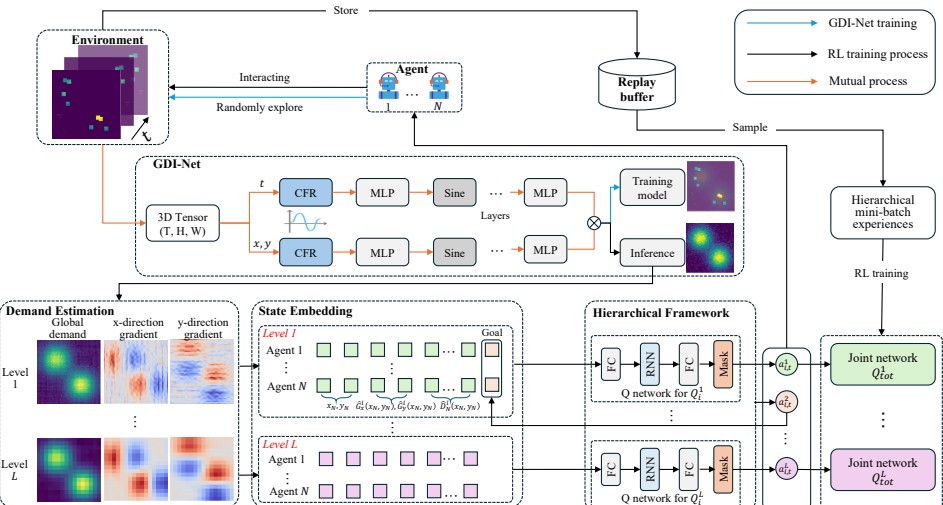

Figure 2: The framework of HGRL. MLP: multi-layer perception. RNN: recurrent neural network.

learning (MARL) framework from a dynamic programming perspective Wang et al. (2024b). Decision-making is decomposed across multiple spatial resolution levels, enabling agents to operate with different planning granularities.

At each hierarchical level $\ell$, the decision process is modeled as a Markov Decision Process (MDP), defined by the tuple $(\mathcal{S}^\ell, \mathcal{A}^\ell, r^\ell, \gamma)$, where $\mathcal{S}^\ell$, $\mathcal{A}^\ell$, and $r^\ell$ denote the state space, action space, and reward function, respectively, and $\gamma$ is the discount factor.

**State:** At each level $\ell$, the state $s_i^\ell \in \mathcal{S}^\ell$ for agent $i$ combines local observations and demand predictions from the GDI-Net.

**Hierarchical Action and Policies.** We design a hierarchical action space across multiple resolution levels. At the lowest level ($\ell = 1$), the action $a_i^1$ selects a specific grid cell within the agent's local $w \times w$ window, directly controlling movement. For higher levels ($\ell > 1$), the action $a_i^\ell$ specifies a coarser region as a sub-goal, guiding subordinate policies toward finer-grained execution. Formally, each policy $\pi_i^\ell$ recursively determines the behavior of its subordinate $\pi_i^{\ell-1}$:

$$\pi_i^\ell(s_i^\ell) = \begin{cases} \text{select fine-scale target,} & \ell = 1, \\ \\ \text{assign sub-goal region to } \pi_i^{\ell-1}, & \ell > 1. \end{cases} \tag{2}$$

**Reward:** High-level policies perform decisions every $\tau$ timesteps, defining sub-goals for lower-level policies. The high-level reward $r_i^\ell$ aggregates lower-level rewards over these intervals:

$$r_i^\ell(t) = \sum_{t'=t}^{t+\tau-1} \gamma^{t'-t} r_i^{\ell-1}(t'). \tag{3}$$

The reward at each timestep combines coverage gain and movement penalties:

$$r_i^\ell = \Delta C(x_{i,t}, y_{i,t}) - \delta \cdot \text{Dist}(x_{i,t}, y_{i,t}, x_{i,t-1}, y_{i,t-1}). \tag{4}$$

Here, $\Delta C(x_{i,t}, y_{i,t}) = C(x_{i,t}, y_{i,t}) - C(x_{i,t-1}, y_{i,t-1})$ measures the coverage increase within the local observation window $a \times a$ (where $C(x_{i,t}, y_{i,t})$ is the current sum coverage of all agents), and $\text{Dist}(\cdot)$ denotes the Euclidean distance moved, penalizing excessive displacement with a unit cost $\delta$.

**Consistency of Hierarchical Reward Objectives**: Our hierarchical reward design maintains consistency and mitigates non-stationarity by ensuring that cumulative rewards at higher levels precisely aggregate rewards at lower levels Wang et al. (2024b). This design ensures that rewards propagate clearly across hierarchical levels, effectively guiding agents toward consistent long-term objectives. Our hierarchical reward design ensures consistency by aggregating lower-level rewards into higher-level returns Wang et al. (2024a), reducing non-stationarity and aligning long-term objectives.

**Proof**: Consider level $\ell$ with decision intervals of length $\tau$. The cumulative reward at this level aggregates lower-level rewards as follows:

$$R_i^\ell(t) = \sum_{t'=t}^{t+\tau-1} \gamma^{t'-t} r_i^{\ell-1}(t') \tag{5}$$

$$= \sum_{t'=t}^{t+\tau-1} \gamma^{t'-t} [\Delta C(x_{i,t'}, y_{i,t'}) - \delta \cdot \mathrm{Dist}(x_{i,t'}, y_{i,t'}, x_{i,t'-1}, y_{i,t'-1})] \tag{6}$$

$$= C(x_{i,t+\tau-1}, y_{i,t+\tau-1}) - C(x_{i,t-1}, y_{i,t-1}) - \delta \sum_{t'=t}^{t+\tau-1} \gamma^{t'-t} \mathrm{Dist}(x_{i,t'}, y_{i,t'}, x_{i,t'-1}, y_{i,t'-1}). \tag{7}$$

This formulation shows that higher-level rewards faithfully summarize lower-level returns, promoting stable and goal-consistent learning across the hierarchy.

### 4.2 MULTI-SCALE STATE EMBEDDING VIA GDI-NET

#### 4.2.1 SPATIOTEMPORAL DEMAND REPRESENTATION AND NETWORK DESIGN

GDI-Net generates structured, multi-resolution representations of spatiotemporal demand. The backbone of GDI-Net is built upon the Spatiotemporal Implicit Neural Representation (ST-INR) model Nie et al. (2024). It takes as input a demand tensor $\mathcal{D} \in \mathbb{R}^{T \times H \times W}$, where each frame $\mathcal{D}_t$ aggregates the local observations of all agents at time $t$. To capture temporal dynamics and spatial semantics Mao et al. (2023), GDI-Net decouples and encodes the temporal and spatial dimensions independently.

**Spatiotemporal Encoding**. To model spatiotemporal demand, GDI-Net separately encodes temporal and spatial coordinates using Fourier features, followed by sine-activated MLPs to capture temporal dynamics and spatial semantics Ji et al. (2020). The resulting embeddings are concatenated to form a unified representation, which is then used to estimate demand intensity $\hat{D}(t, x, y)$ and reshaped into a tensor of shape $[T, H, W]$.

**Multi-Resolution Demand Modeling.** To support hierarchical decision-making, GDI-Net produces demand maps at multiple spatial resolutions. Specifically, for each level $\ell$, we obtain a coarse-grained map $\hat{D}^\ell(t, x, y)$ via resolution-specific spatial pooling applied to the base resolution output. Formally,

$$\hat{D}^\ell(t, x, y) = \mathrm{Pooling}_\ell(\hat{D}(t, x', y')), \quad (x', y') \in \Omega_\ell(x, y)$$

where $\Omega_\ell(x, y)$ denotes the receptive field for cell $(x, y)$ at level $\ell$.

To further guide directional planning, Sobel filters are applied to compute gradient maps $\hat{G}_x^\ell$ and $\hat{G}_y^\ell$, capturing horizontal and vertical demand trends Jiang et al. (2024). These multi-scale representations $\{\hat{D}^\ell, \hat{G}_x^\ell, \hat{G}_y^\ell\}_{\ell=1}^L$ provide each policy layer with resolution-aligned, demand-aware inputs for coarse-to-fine planning.

#### 4.2.2 MULTI-SCALE ALIGNMENT AND STATE EMBEDDING

A core feature of our framework is the structured alignment between GDI-Net's multi-resolution outputs and the hierarchical policy architecture Zhang et al. (2017b). To our knowledge, this is the first attempt to explicitly align multi-scale demand perception with decision layers in hierarchical MARL.

At each level $\ell$, GDI-Net outputs demand intensity $\hat{D}^\ell$ and gradient fields $\hat{G}_x^\ell$, $\hat{G}_y^\ell$ over a spatial grid $\mathcal{G}^\ell$, matching the granularity at which the policy $\pi^\ell$ operates. This ensures that agents receive semantically consistent information across decision scales Cai et al. (2024).

To generate coarse-level maps, GDI-Net applies average pooling over finer-scale regions:

$$\hat{D}^{\ell+1}(t, x, y) = \frac{1}{|\Omega(x, y)|} \sum_{(x', y') \in \Omega(x, y)} \hat{D}^\ell(t, x', y'), \quad (x, y) \in \mathcal{G}^{\ell+1}, \tag{8}$$

with spatial gradients approximated accordingly:

$$\hat{G}x,y^{\ell+1}(t,x,y) \approx \frac{1}{|\Omega(x,y)|} \sum (x',y') \in \Omega(x,y)\hat{G}^\ell_{x,y}(t,x',y'). \tag{9}$$

This design guarantees that directional trends and demand densities remain consistent across layers. High-level policies produce sub-goals over $\hat{D}^{\ell+1}$, while lower-level agents operate using $\hat{D}^\ell$ and follow refined gradient cuesHu et al. (2018). This results in an approximate consistency of expected demand values under hierarchical policies:

$$\mathbb{E}(x,y) \sim \pi^{\ell+1}\left[\hat{D}^{\ell+1}(t,x,y)\right] \approx \mathbb{E}(x',y') \sim \pi^\ell\left[\hat{D}^\ell(t,x',y')\right]. \tag{10}$$

**State Embedding.** Based on the aligned multi-resolution perception, each agent $i$ at level $\ell$ constructs its state embedding from its current position $(x_i, y_i)$, local demand and gradient values, and the sub-goal $a_i^{\ell+1}$ assigned by the higher-level policy:

$$s_i^\ell = \left(x_i, y_i, \hat{D}^\ell(x_i, y_i), \hat{G}_x^\ell(x_i, y_i), \hat{G}_y^\ell(x_i, y_i), a_i^{\ell+1}\right). \tag{11}$$

Here, $a_i^{\ell+1} \in \mathcal{G}^\ell$ denotes the spatial sub-goal output from the upper-level policy $\pi_i^{\ell+1}$, which serves as a goal constraint or guidance for decision-making at level $\ell$. This embedding combines perceptual signals with hierarchical intent, enabling goal-conditioned control under partial observability Ji et al. (2019).

More detailed information about GDI-Net can be founded in Appendix B

### 4.3 Hierarchical Reinforcement Learning with Value Decomposition Networks

Our method employs a hierarchical reinforcement learning structure, integrating multi-scale state embeddings provided by GDI-Net with a layered value decomposition strategy inspired by Value Decomposition Networks (VDN) Sunehag et al. (2018). At each hierarchical level $\ell$, agents make decisions based on local state embeddings $s_i^\ell$, which combine immediate local observations and GDI-Net's level-specific demand and gradient features.

At each level $\ell$, agent $i$ computes local Q-values via an FC-RNN-FC network:

$$Q_i^\ell(s_i^\ell, a_i^\ell; \theta_i^\ell), \quad \text{for each agent } i. \tag{12}$$

Per-agent values are linearly combined into a joint Q-value for level $\ell$:

$$Q_{\text{tot}}^\ell(\mathbf{s}^\ell, \mathbf{a}^\ell; \theta^\ell) = \sum_i Q_i^\ell(s_i^\ell, a_i^\ell; \theta_i^\ell), \tag{13}$$

where $\mathbf{s}^\ell$ and $\mathbf{a}^\ell$ denote the joint state and action vectors at hierarchical level $\ell$, respectively.

### 4.4 Training

Our hierarchical reinforcement learning system jointly trains GDI-Net and hierarchical VDN networks in an iterative, closed-loop manner. The training loop alternates between supervised demand reconstruction and reinforcement learning updates Zhang et al. (2014), leveraging mutual information sharing between the two modules to enhance both demand modeling accuracy and policy effectiveness.

**Supervised Training of GDI-Net.** GDI-Net is trained to reconstruct global demand fields from aggregated local observations collected during agent interactions. The objective is to minimize the mean squared error (MSE) between predicted and observed demands:

$$\mathcal{L}_{\text{GDI}}(\phi) = \frac{1}{|\mathcal{D}|} \sum_{(t,x,y)\in\mathcal{D}} \left(D(t,x,y) - \hat{D}(t,x,y;\phi)\right)^2, \tag{14}$$

where $D(t,x,y)$ represents the true observed demand at time $t$ and location $(x,y)$, and $\hat{D}(t,x,y;\phi)$ is the GDI-Net prediction parameterized by $\phi$. This supervised training ensures that GDI-Net provides

accurate multi-scale spatiotemporal demand estimations to guide the hierarchical policies Zhang et al. (2015).

**Hierarchical VDN Training.** Agents at each hierarchy level $\ell$ independently compute their local Q-values $Q_i^\ell(s_i^\ell, a_i^\ell; \theta_i^\ell)$, which are then linearly aggregated into joint Q-values via Value Decomposition Networks (VDN):

$$Q_{\text{tot}}^\ell(\mathbf{s}^\ell, \mathbf{a}^\ell; \theta^\ell) = \sum_i Q_i^\ell(s_i^\ell, a_i^\ell; \theta_i^\ell). \tag{15}$$

Training is performed under the centralized training decentralized execution (CTDE) paradigm, utilizing the VDN temporal-difference (TD) loss function:

$$\mathcal{L}_{\text{VDN}}(\theta^\ell) = \mathbb{E}\left[(r^\ell + \gamma \max_{\mathbf{a}'} Q_{\text{tot}}^\ell(\mathbf{s}', \mathbf{a}'; \theta^{\ell-}) - Q_{\text{tot}}^\ell(\mathbf{s}, \mathbf{a}; \theta^\ell))^2\right], \tag{16}$$

where $r^\ell$ is the aggregated hierarchical reward, $\gamma$ is the discount factor, and $\theta^{\ell-}$ denotes periodically synchronized target network parameters.

**Closed-Loop Joint Optimization.** The system operates in a feedback loop where agents explore the environment and store experiences in a shared replay buffer Zhang et al. (2021). These samples are used to jointly update GDI-Net and the hierarchical VDN policies. Specifically, GDI-Net learns demand representations from trajectories, enhancing state embeddings, while hierarchical policies leverage these embeddings to make better decisions Pan et al. (2019a). This forms a synergistic cycle: improved demand modeling guides policy learning, and refined policies generate better data for representation learning.

## 5 EXPERIMENTS

### 5.1 EXPERIMENTAL SETTING

**Datasets.** We evaluate our method on the following two datasets:

- **Toy Example:** A synthetic spatiotemporal dataset defined on a $50 \times 50$ grid. At each time step, the demand map is formed by two Gaussian peaks centered at $(15, 15)$ and $(35, 35)$, with shared standard deviation $\sigma = 0.15$, i.e., $D(x, y) = \exp(-\frac{(x-\mu_x)^2 + (y-\mu_y)^2}{2\sigma^2})$. The final demand is the sum of the two Gaussians, normalized to $[0, 1]$, and perturbed with Gaussian noise $\varepsilon(x, y) \sim \mathcal{N}(0, (0.05 \cdot D_{\max})^2)$. The dataset contains $T = 28$ timesteps, resulting in a 3D tensor of shape $[28, 50, 50]$.

- **TaxiBJ Zhang et al. (2017a):** A real-world dataset of taxi trajectories in Beijing. We use the first 7 days of data, divided into 48 intervals per day (30 minutes each), resulting in $T = 336$ steps. The spatial grid is resized to $64 \times 64$, forming a 3D tensor of shape $[336, 64, 64]$.

For the training of GDI-Net, demand observations are collected from agent trajectories via random exploration, and the entire dataset is used without an explicit train-validation split due to the non-stationary nature of the generation process. For reinforcement learning training, we partition the dataset into training and evaluation sets using a 6:1 ratio to assess the model's generalization to unseen demand patterns.

**Evaluation Metrics and Implementation.** We evaluate our framework using three key metrics. **Average Coverage (AC)** measures the proportion of demand successfully covered by agents within their local windows over time, while **Average Transportation Distance (ATD)** quantifies the average Manhattan distance traveled by agents, reflecting movement efficiency. A composite **Score** balances AC and ATD through a trade-off coefficient of 0.5, capturing overall system performance. Detailed definitions, formulas, hyperparameters, and evaluation setups are provided in Appendix C.

**Baselines.** We compare our method against three categories of baselines. Heuristics: **EADS** Ruan et al. (2020), a rule-based assignment method without learning or adaptation. **Classic MARL:** **VDN** Sunehag et al. (2018), which linearly decomposes Q-values, and **QMIX** Rashid et al. (2018), which introduces a monotonic mixing network for coordinated value estimation. **Structured RL:** **GridNet** Han et al. (2019), **HMF** Yu (2023), **HMF (IPC)** Yu (2023), and **MDPO** Ma et al. (2024), representing approaches that enhance coordination through spatial encoders, hierarchical decomposition, dynamic grouping, or communication mechanisms. For fair comparison, all baselines adopt the

Table 1: Model comparison on the **Toy Example** and **TaxiBJ** datasets. Metrics where higher values are better are marked with ↑, others with ↓. Bold and underlined values indicate the best and second-best results.

| Datasets | Category | Algorithm | AC ↑ | ATD ↓ | Score ↑ | Training Time ↓ | Planning Time ↓ |
|---|---|---|---|---|---|---|---|
| Toy Example | Classic MARL | VDN | 5.782 | 4.55 | 0.333 | 158.29 | 0.522 |
| | | QMIX | 4.1582 | 4.68 | 0.178 | 165.44 | 0.684 |
| | | MAPPO | 1.19 | 6.05 | -0.154 | 162.77 | 0.560 |
| | | QPLEX | 3.52 | 6.44 | 0.045 | 169.81 | 0.712 |
| | Structured RL | GridNet | 4.262 | 11.31 | -0.092 | 132.36 | 1.318 |
| | | HMF | 4.22 | 14.27 | -0.220 | 189.52 | 0.714 |
| | | HMF (w/ IPC) | 4.34 | 13.38 | -0.171 | 187.73 | 0.713 |
| | | MDPO | 5.82 | 13.81 | -0.053 | 1075 | 0.740 |
| | Heuristics | EADS | 5.025 | **2.38** | 0.354 | N.A. | 2.03 |
| | HRL + Gradient | HGRL (ours) | **6.601** | 3.81 | **0.440** | 195.4 | 0.793 |
| TaxiBJ | Classic MARL | VDN | 477.5813 | 0.0979 | 0.119 | 166.13 | 0.708 |
| | | QMIX | 619.9771 | 0.4479 | 0.206 | 167.08 | 0.797 |
| | | MAPPO | 445.4 | 0.1479 | 0.095 | 173.64 | 0.755 |
| | | QPLEX | 304.9 | **0.0917** | 0.000 | 171.32 | 0.750 |
| | Structured RL | GridNet | 620.5 | 3.5200 | 0.106 | 145 | 15.82 |
| | | HMF | 605.3 | 15.4400 | -0.293 | 299.71 | 8.544 |
| | | HMF (w/ IPC) | 533.2 | 8.1400 | -0.105 | 248.1 | 8.283 |
| | | MDPO | 898.5 | 3.9700 | 0.283 | 1122 | 9.312 |
| | Heuristics | EADS | 664 | 0.3100 | 0.241 | N.A. | 104 |
| | HRL + Gradient | HGRL (ours) | **1030** | 0.2000 | **0.496** | 209.3 | 10.25 |

same GDI-Net inputs where applicable, excluding the multi-scale gradient information unique to our method. More details can be founded in Appendix D.

## 5.2 MODEL COMPARISON

We evaluate four categories of baselines on both the **Toy Example** and **TaxiBJ**: **Classic MARL** (VDN, QMIX, MAPPO, QPLEX), **Structured RL** (GridNet, HMF, HMF w. IPC, MDPO), **Heuristics** (EADS), and our method **HGRL**. Results are summarized in Table 1.

On the **Toy Example**, HGRL attains the best AC and Score (AC: **6.601**, Score: 0.440), exceeding the strongest alternatives (MDPO, AC: 5.82; EADS, Score: 0.354). EADS achieves the lowest ATD (2.38) with VDN second (4.55), while HGRL remains competitive (3.81) and delivers the highest overall Score. In efficiency, GridNet trains fastest (132.36 mins) and VDN plans fastest (0.522 s); HGRL's training (195.4 mins) and planning (0.793 s) are moderate given its performance gains.

On the **TaxiBJ**, HGRL again leads on effectiveness, achieving the highest AC and Score (AC: 1030, Score: 0.496) versus the next-best MDPO (AC: 898.5, Score: 0.283) and EADS (Score: 0.241). Classic MARL yields the smallest ATD (QPLEX: 0.0917, VDN: 0.0979), but their AC/Score are clearly lower. Structured RL variants are competitive on AC but incur high computation (e.g., MDPO training: 1122 mins; HMF planning: 8.544 s). HGRL's training time is moderate (209.3 mins); its planning time (10.25 s) is higher than value-based MARL, reflecting the cost of hierarchical lookahead that yields the top AC/Score.

Across both settings, HGRL provides the best overall effectiveness with highest AC and Score while maintaining acceptable efficiency, whereas EADS excels in ATD and Classic MARL in planning latency. The results indicate that hierarchical, gradient-informed coordination offers a superior trade-off between coverage quality and movement efficiency in both synthetic and real-world environments.

## 5.3 ABLATION STUDY

**Ablation on HG and GDI-Net.** Figure 3 evaluates four variants: (i) w/ HG, w/ GDI-Net (full model), (ii) w/ HG, w/o GDI-Net (hierarchical RL without global demand estimation), (iii) w/ G, w/ GDI-Net (single-scale gradient-informed RL with GDI-Net), and (iv) w/o HG, w/o GDI-Net (baseline without both modules) on **TaxiBJ**. As the figure shows, incorporating HG markedly enhances scalability: with increasing numbers of agents, variants with HG achieve higher AC and converge more stably, whereas single-scale counterparts (G) suffer from lower rewards and instability. GDI-Net, on the other hand, alleviates partial observability by providing reliable global demand estimation, leading to consistently reduced ATD. The full model that integrates both HG and GDI-Net

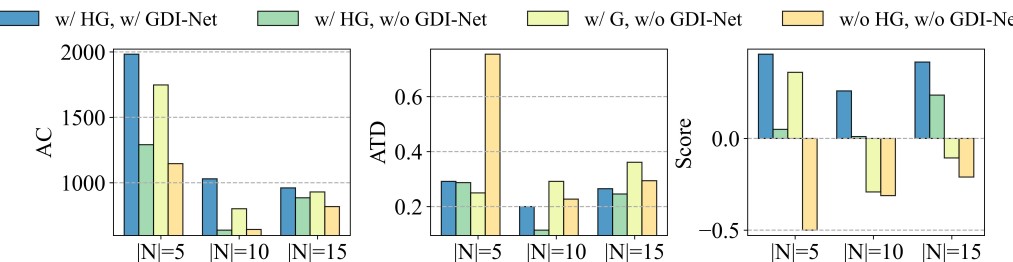

Figure 3: Ablation study (TaxiBJ) on the contributions of the hierarchical gradient (HG) module and the global demand inference network (GDI-Net) under varying numbers of agents ($|N| = 5, 10, 15$). Results are reported in terms of **AC**, **ATD**, and overall **Score**.

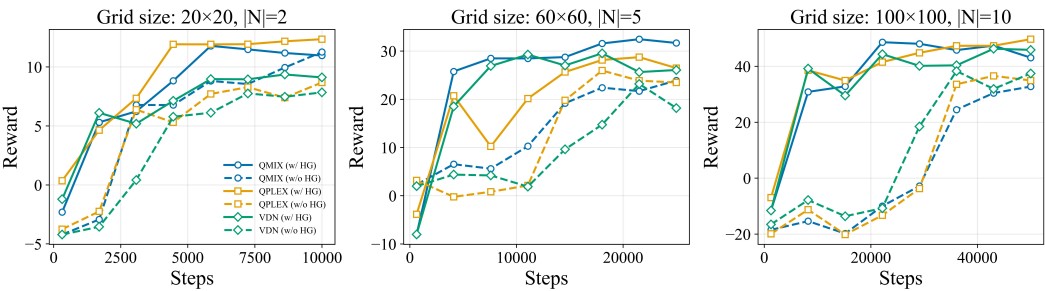

Figure 4: Scalability study of classic MARL methods (QMIX, QPLEX, and VDN) with and without the hierarchical gradient-informed RL module (HG) under different grid sizes on **Toy Example**.

attains the best overall Score across all settings, while removing either module results in notable performance degradation. These findings confirm that (i) hierarchical multi-scale design is essential for scalability, and (ii) global demand inference is critical for efficient coordination under partial observability.

**Impact of Hierarchical Gradient Module on Scalability.** Figure 4 compares classic MARL baselines (QMIX, QPLEX, and VDN) with and without the hierarchical gradient (HG) module under increasing grid sizes and agent numbers in a deterministic environment, where the training steps are dynamically adjusted to the problem scale. While the performance gap between w/ HG and w/o HG remains small in the 20×20 setting, the advantage of HG becomes increasingly evident as the scale grows. On medium (60×60) and large (100×100) grids, methods equipped with HG converge faster, achieve higher rewards, and maintain greater training stability. In contrast, their w/o HG counterparts converge slowly and suffer severe performance degradation. These results underscore the critical role of HG in enabling MARL methods to scale effectively to more complex and realistic scenarios.

**More details.** Mode details in visulization results of GDI-Net can be founded in Appendix E

## 6 CONCLUSION

In this paper, we proposed **HGRL**, a hierarchical gradient-informed RL framework integrated with a GDI-Net model to address DRAPs. To tackle the scalability issue, our framework decomposes decision-making into multiple spatial scales, significantly reducing complexity. For partial observability, GDI-Net infers multi-resolution global demand and directional gradients from local observations, enriching state representations and enhancing agent exploration efficiency. Experimental results on both synthetic and real-world datasets validate that our approach consistently outperforms heuristic, state-of-the-art MARL and structured RL baselines in terms of demand coverage and overall resource allocation effectiveness. Our hierarchical design and gradient-informed exploration strategy provide a principled solution to large-scale DRAPs, promising broader applicability in complex, real-time decision-making scenarios.

ETHICS STATEMENT

This work adheres to the ICLR Code of Ethics. Our study does not involve human subjects, sensitive personal data, or potentially harmful applications. All datasets used are publicly available and properly cited, and no privacy or security issues are introduced. The proposed methods are intended purely for academic research, with potential applications in sustainable transportation and logistics. We believe our work raises no ethical concerns.

REPRODUCIBILITY STATEMENT

We have made significant efforts to ensure reproducibility. The problem formulation, models, and algorithms are fully described in Section 3 and Section 4. Detailed training setups, hyperparameters, and evaluation metrics are provided in the Appendix C. All proofs and derivations are included in main paper and Appendix A. An anonymous implementation of our framework, along with processed datasets and experiment scripts, is available at: `https://anonymous.4open.science/r/HGRL4DRA-B40FS/`.

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

## A  PROOF OF NP-HARDNESS

**Theorem:** The dynamic resource allocation problem, as defined in Section 3, is NP-hard.

**Proof:** We establish the result via a polynomial-time reduction from the classical Maximum Coverage Problem (MCP) Feige (1998), which is known to be NP-hard.

**Maximum Coverage Problem (MCP).** Given a universe of elements $U = \{e_1, e_2, \ldots, e_m\}$ and a collection of subsets $S = \{S_1, S_2, \ldots, S_n\}$ with each $S_i \subseteq U$, the goal is to select at most $k$ subsets to maximize the number of covered elements.

To reduce MCP to our setting, we construct an instance of our problem with a single time period ($|T| = 1$). Each element $e_j \in U$ is mapped to a grid cell $g_j \in G$ with unit demand $D_{1,g_j} = 1$, while all other cells have zero demand. We define one resource ($|N| = 1$) with capacity $C_n \geq m$ and a service window large enough to cover any subset $S_i$. Each subset $S_i$ corresponds to a feasible location such that placing the resource there will cover the grid cells representing elements in $S_i$. We set the movement penalty parameter $\delta = 0$ to eliminate any dynamic cost, reducing the objective to maximizing total served demand.

In this construction, selecting $k$ subsets in MCP is equivalent to choosing $k$ resource placements in our problem to cover maximal demand. Since this reduction is polynomial-time, and MCP is NP-hard, our problem is also NP-hard.

**Remark:** While the reduction uses a simplified single-period setting, our full dynamic formulation introduces temporal coupling and routing costs across multiple periods. This makes the problem strictly more complex than static MCP, incorporating characteristics of sequential decision making and vehicle routing under spatial-temporal constraints.

# B    DETAILED STRUCTURE OF GDI-NET

We provide a detailed description of GDI-Net in this section, as introduced in Section 4.2, including its input encoding, architectural components, and multi-scale output design.

## B.1    INPUT REPRESENTATION VIA CONCATENATED RANDOM FOURIER FEATURES (CRF)

Our GDI-Net employs Concatenated Random Fourier Features (CRF) to effectively encode continuous spatiotemporal coordinates $(t, x, y)$, mapping raw coordinates into high-dimensional embeddings capable of capturing intricate spatiotemporal dependencies.

Given an input tensor of dimensions $(T, H, W)$, representing temporal and spatial dimensions, we separately encode temporal and spatial coordinates using CRF, followed by dedicated neural layers with sine activations.

**Temporal Encoding:** The temporal coordinate $t \in [0, T]$ is first encoded by random Fourier features:

$$\phi_t(t) = [\sin(2\pi \mathbf{B}_t t), \cos(2\pi \mathbf{B}_t t)], \tag{17}$$

where $\mathbf{B}_t \in \mathbb{R}^d$ is a randomly initialized, trainable frequency matrix. The encoded representation passes through multiple layers, each composed of a linear MLP followed by a sine activation function:

$$z_t(t) = \text{MLP}_t^{(L)}(\sigma(\ldots \sigma(\text{MLP}_t^{(1)}(\phi_t(t))))), \tag{18}$$

where $\sigma(\cdot) = \sin(\cdot)$ denotes the sine activation function.

**Spatial Encoding:** Similarly, spatial coordinates $(x, y) \in [0, H] \times [0, W]$ are encoded using CRF:

$$\phi_{x,y}(x, y) = [\sin(2\pi \mathbf{B}_x x), \cos(2\pi \mathbf{B}_x x), \sin(2\pi \mathbf{B}_y y), \cos(2\pi \mathbf{B}_y y)], \tag{19}$$

where $\mathbf{B}_x, \mathbf{B}_y \in \mathbb{R}^d$ are similarly random frequency matrices. This representation also passes through a series of layers consisting of linear transformations followed by sine activations:

$$z_s(x, y) = \text{MLP}_s^{(L)}(\sigma(\ldots \sigma(\text{MLP}_s^{(1)}(\phi_{x,y}(x, y))))). \tag{20}$$

## B.2    NETWORK ARCHITECTURE AND FUSION

After independently encoding the temporal and spatial coordinates, their resulting embeddings $z_t(t)$ and $z_s(x, y)$ are concatenated:

$$z(t, x, y) = [z_t(t); z_s(x, y)]. \tag{21}$$

A subsequent fusion MLP processes the combined embedding to infer the demand intensity at each spatiotemporal coordinate:

$$\hat{D}(t, x, y) = \text{MLP}_f(z(t, x, y)). \tag{22}$$

The final output tensor, reshaped to $(T, H, W)$, represents the predicted spatiotemporal demand intensity across the domain.

## B.3    MULTI-SCALE RESOLUTIONS HIERARCHICAL OUTPUTS

To facilitate hierarchical decision-making, GDI-Net generates multiple resolution outputs through spatial aggregation. Demand maps at coarser scales are computed via average pooling from higher-resolution predictions:

$$\hat{D}^{\ell+1}(t, x, y) = \frac{1}{|\Omega(x, y)|} \sum_{(x', y') \in \Omega(x, y)} \hat{D}^\ell(t, x', y'), \tag{23}$$

where $\Omega(x, y)$ denotes the finer-scale spatial cells aggregated into coarser grid cell $(x, y)$ at level $\ell + 1$.

### B.4 GRADIENT COMPUTATION USING SOBEL FILTERS

After generating multi-resolution demand maps, we compute spatial gradients using fixed 2D Sobel filters:

$$K_x = \begin{bmatrix} -1 & 0 & 1 \\ -2 & 0 & 2 \\ -1 & 0 & 1 \end{bmatrix}, \quad K_y = \begin{bmatrix} -1 & -2 & -1 \\ 0 & 0 & 0 \\ 1 & 2 & 1 \end{bmatrix}. \tag{24}$$

Horizontal and vertical gradients are computed through convolution:

$$\hat{G}_x^\ell = \text{Conv2d}(\hat{D}^\ell, K_x), \quad \hat{G}_y^\ell = \text{Conv2d}(\hat{D}^\ell, K_y). \tag{25}$$

These gradients provide directional guidance, steering agents toward areas of significant demand variation, thereby enhancing decision-making efficiency in sparse-reward environments.

## C EVALUATION METRICS AND IMPLEMENTATION

Three metrics used to validate our methods are defined as follows:

$$\text{AC} = \frac{1}{T} \sum_{t=1}^{T} \frac{\sum_i \sum_{(x,y) \in W_i} D(t,x,y)}{\sum_{x,y} D(t,x,y)}, \tag{26}$$

$$\text{ATD} = \frac{1}{NT} \sum_{t=1}^{T} \sum_{i=1}^{N} \left( |x_{i,t} - x_{i,t-1}| + |y_{i,t} - y_{i,t-1}| \right), \tag{27}$$

$$\text{Score} = 0.5 \cdot \tilde{\text{AC}} - 0.5 \cdot \tilde{\text{ATD}}, \tag{28}$$

where $W_i$ denotes the coverage window of agent $i$, $\delta$ is a weighting coefficient for the distance penalty, and $\tilde{\text{AC}}$ and $\tilde{\text{ATD}}$ represent the scaled values of AC and ATD, respectively. The window size of local view is set to $3 \times 3$

All experiments were implemented in Python using PyTorch. Training was conducted on a server with an NVIDIA A6000 GPU and AMD 7763 CPU.

**Training Parameters.** We used the RMS optimizer with an initial learning rate of $1 \times 10^{-4}$ for both GDI-Net and all hierarchical Q-networks. Learning rate decay was scheduled based on validation performance. Each model was trained for 500k steps. Batch size was set to 32, and replay buffer size was 20,000. Target networks were updated every 200 steps.

**Exploration Strategy.** Agents followed an $\epsilon$-greedy policy during training, with $\epsilon$ linearly annealed from 1.0 to 0.05 over the first 150k steps.

**GDI-Net Specifics.** The MLPs in GDI-Net consist of 3 hidden layers with 128 units each and sine activations. The Fourier encoding dimension was set to $d = 2048$. For multi-resolution output, we used three spatial scale of resolutions with downsampling ratios of 1, 1/4, and 1/16.

**Hierarchical RL Architecture.** Each policy level employs a three-layer FC-RNN-FC structure. Specifically, the input layer receives the state embedding, followed by a GRU layer with a hidden size of 64, and finally an output layer whose dimension depends on the action space at each hierarchical level. We set the number of hierarchy levels to $L = 2$: the first policy level aligns with spatial resolutions 1 and 2 from GDI-Net, while the second policy level aligns with spatial resolutions 2 and 3. Additionally, we set the decision interval $\tau$ between adjacent policy levels to 3 environment steps.

**Evaluation Settings.** All evaluation results are averaged over 3 random seeds. The trade-off coefficient $\delta$ in the Score metric is set to 0.2 for the **Toy Example** and 0.75 for the **TaxiBJ** dataset. Unless otherwise specified, we use $N = 10$ agents and simulate $T = 20$ timesteps per episode.

**Code and Reproducibility.** To support reproducibility, we provide open access to our codebase and configuration files at `https://anonymous.4open.science/r/HGRL4DRA-B40FS/`, including instructions for reproducing the main experimental results.

## D    BASELINES DESCRIPTION

For clarity, we provide additional descriptions of the baselines used in our experiments.

*Classic MARL:* **VDN** Sunehag et al. (2018) serves as both a baseline and the foundation of our hierarchical RL framework. It aggregates individual Q-values linearly but operates only on single-scale demand inputs without hierarchical abstraction. **QMIX** Rashid et al. (2018) extends VDN by introducing a learned monotonic mixing network for coordinated value estimation, albeit with increased training complexity.

*Structured RL:* **GridNet** Han et al. (2019) implements a grid-wise spatial encoder-decoder policy network. The encoder processes the global demand map with feature dimension $\phi_c = 1$ (demand per grid), and the decoder outputs per-grid action distributions with dimension $\phi_a = w^2$, where $w$ is the local window size. Agent coordination emerges implicitly through the large receptive field of stacked convolutional layers, enabling spatially coherent decision-making. **HMF** Yu (2023) is a hierarchical mean-field framework that approximates the global joint Q-function via a two-level decomposition. The *low-level* uses attention-based mean-field Q-networks ($Q_{\text{LOC}}$, $Q_{\text{GMF}}$) for intra-group coordination, while the *high-level* supervisor Q-network ($Q_{\text{TOT}}$) coordinates across groups. We adopt 3 agent groups in our experiments. **HMF (w/ IPC)** Yu (2023) extends HMF with dynamic grouping via Information and Policy Consistency (IPC), periodically reassigning agents to groups based on policy similarity to enhance coordination adaptively. **MDPO** Ma et al. (2024) combines decentralized execution with networked neighbor communication, originally for partially observable graph-based environments with hop-$k$ message passing. Since our setting lacks a fixed graph, we model agents as a fully connected graph with $k = 1$, enabling unrestricted communication.

*Heuristics:* **EADS** Ruan et al. (2020) is a rule-based method that assigns agents to high-demand regions without learning or adaptive coordination.

For fair comparison, all baselines use the same GDI-Net inputs where applicable, excluding the multi-scale gradient information unique to our method.

## E    MORE DETAILS IN EXPERIMENTS

The results in Figure 6 demonstrate that GDI-Net can accurately reconstruct spatiotemporal demand distributions across multiple resolutions, even when supervised only with sparse agent-collected observations. In both the **Toy Example** and **TaxiBJ** datasets, the estimated demand maps closely match the ground truth, preserving key spatial structures. Furthermore, the derived gradient fields capture meaningful directional patterns, which can effectively guide agent movement. Notably, GDI-Net maintains robust performance even at coarse resolutions (e.g., $4 \times 4$), highlighting its ability to infer global structure from limited inputs.

As shown in Table 2, estimation errors decrease consistently from high to low resolutions across both datasets. This indicates that GDI-Net captures coarse-grained demand patterns more reliably, and the multi-scale structure facilitates robust reconstruction even under sparse supervision. The overall performance on the real-world **TaxiBJ** dataset further demonstrates the model's generalizability.

These results highlight two key advantages of GDI-Net. First, the multi-scale design improves robustness: lower-resolution outputs are less sensitive to noise and data sparsity, enabling more stable recovery of global demand structure. Second, the model generalizes well despite being trained on partial, agent-collected trajectories, accurately reconstructing demand fields and gradients. This suggests that GDI-Net can serve as a reliable perceptual backbone in reinforcement learning systems operating under partial observability.

Figure 5 shows that GDI-Net produces coherent gradient fields across multiple spatial scales, with vectors consistently pointing toward high-demand areas. These gradients act as soft heuristics that guide RL agents under partial observability, enabling hierarchical path planning through coarse-to-fine directional cues. However, in the **TaxiBJ** dataset, gradient accuracy degrades at very low resolutions due to the disproportionate size of the Sobel kernel relative to the grid. This highlights the need for scale-aware or learnable alternatives in future work.

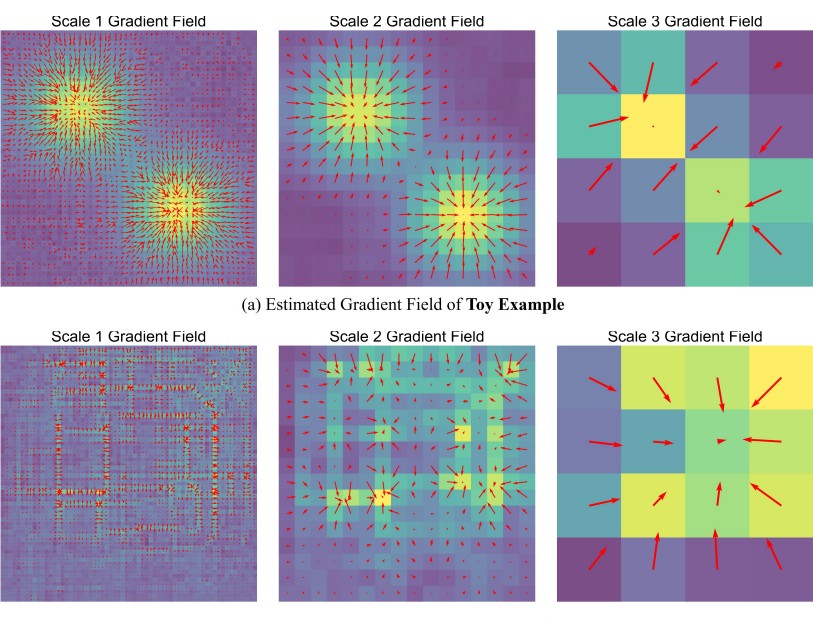

(a) Estimated Gradient Field of **Toy Example**

(b) Estimated Gradient Field of **TaxiBJ**

Figure 5: **Estimated gradient fields across multiple spatial scales.** (a) Toy Example. (b) TaxiBJ dataset. Each figure shows the reconstructed demand intensity overlaid with gradient vectors ($\hat{G}_x^\ell$, $\hat{G}_y^\ell$) estimated via Sobel filters. Arrows indicate the direction of demand increase. Across scales, GDI-Net captures coherent directional structures, allowing coarse-to-fine navigation under partial observability.

Table 2: Estimation errors at different scales for each dataset.

| Dataset | Scale | RMSE | Mean Error | Max Error |
|---|---|---|---|---|
| **Toy Example** | Scale 1 | 0.0593 | 0.0465 | 0.2324 |
| | Scale 2 | 0.0265 | 0.0201 | 0.0954 |
| | Scale 3 | 0.014 | 0.0113 | 0.0344 |
| **TaxiBJ** | Scale 1 | 39.04 | 27.79 | 313.1 |
| | Scale 2 | 17.87 | 13.73 | 55.3 |
| | Scale 3 | 9.932 | 7.23 | 21.78 |

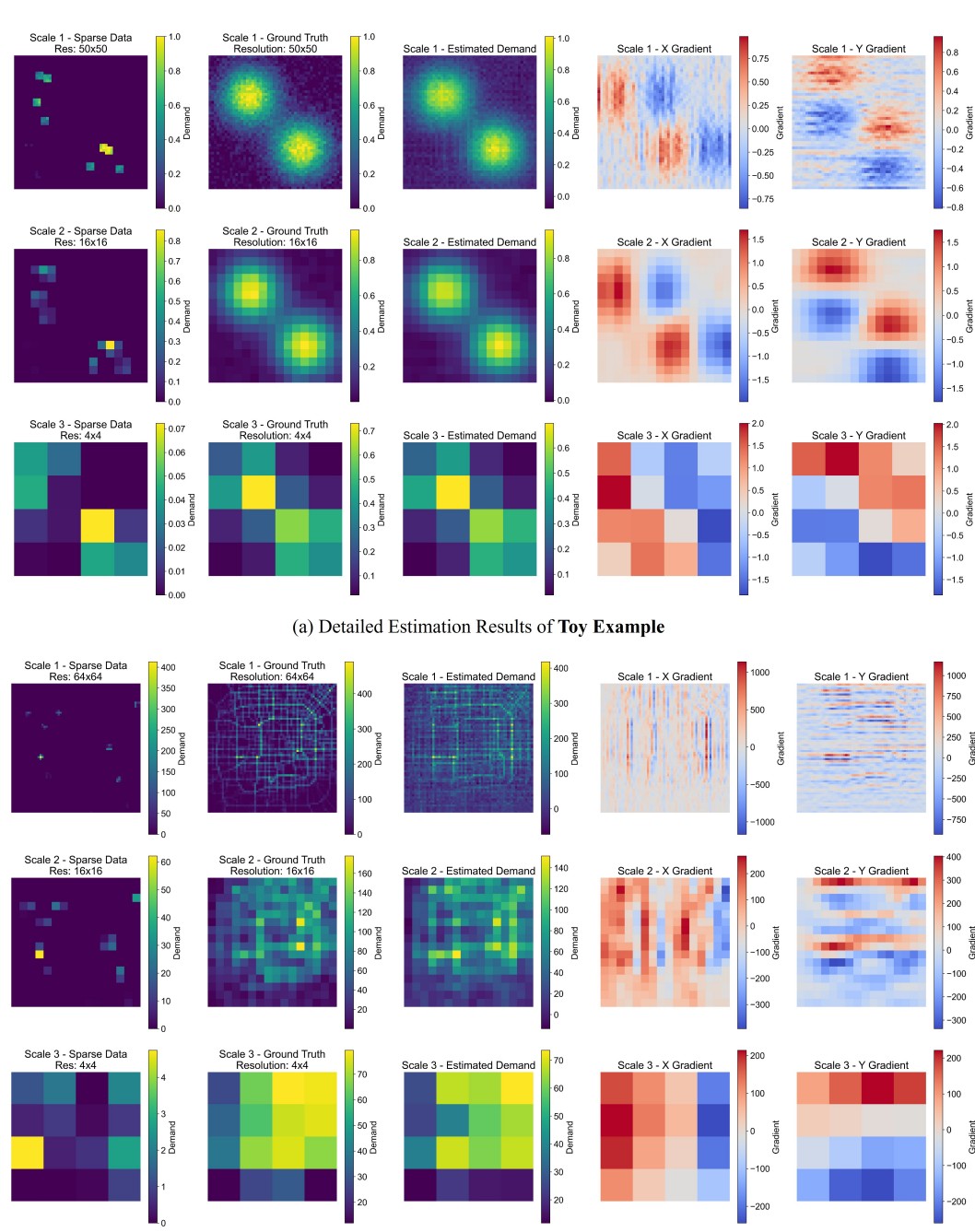

(a) Detailed Estimation Results of **Toy Example**

(b) Detailed Estimation Results of **TaxiBJ**

Figure 6: **Multi-scale demand and gradient estimation results of GDI-Net.** (a) **Toy Example.** (b) **TaxiBJ Dataset.** For each scale (from top to bottom), we visualize: sparse observations collected from RL agent–environment interactions, ground-truth demand, GDI-Net's estimated demand, and the corresponding X/Y spatial gradients computed using Sobel filters. The sparse input data is generated by collecting partial state observations during early-stage random exploration of the environment. Despite the limited and non-uniform coverage, GDI-Net is able to reconstruct coherent demand maps across scales and compute meaningful gradient fields that guide agent movement.

## F    LIMITATIONS AND FUTURE WORK

While the proposed hierarchical structure is well-suited for demand-based DRAPs, our current focus remains on this specific class of problems. In future work, we aim to extend the framework to broader DRAP settings, such as large-scale taxi dispatching and warehouse robot scheduling, thereby exploring its generality in diverse real-world domains.

## G    USE OF LARGE LANGUAGE MODELS

LLMs were used solely as assistive tools for language polishing and LaTeX formatting. They did not contribute to research ideation, modeling, experimentation, or result interpretation. All conceptual and technical contributions are original to the authors.

For transparency, typical usage included prompts such as: "Refine the following paragraph for clarity and conciseness in academic writing" and "Convert this plain text into LaTeX tabular format with proper alignment".

These interactions were limited to improving readability and presentation, without generating new scientific content.

