# OpenReview forum: "Hierarchical Gradient-Informed Reinforcement Learning for Scalable and Partially Observable Dynamic Resource Allocation"
_ICLR.cc/2026/Conference — ICLR 2026 Conference Withdrawn Submission_

### Official Review · Reviewer_NtHm · 2025-10-28

**Soundness:** 3
**Presentation:** 3
**Contribution:** 2
**Rating:** 4
**Confidence:** 5

**Summary:**

This paper addresses dynamic resource allocation problems (DRAPs) in domains such as transportation and energy management. The authors propose a Hierarchical Gradient-Informed Reinforcement Learning (HGRL) framework that combines hierarchical multi-agent reinforcement learning with a Global Demand Inference Network (GDI-Net). The approach decomposes DRAPs into multi-scale subproblems to improve scalability and leverages GDI-Net to infer global demand gradients from local observations, mitigating partial observability. Empirical results on synthetic and real-world (ride-hailing) datasets indicate substantial performance improvements over baseline methods in demand coverage and transportation efficiency.

**Strengths:**

1. Methodological clarity: The overall framework is clearly described and conceptually coherent. The hierarchical design and integration of demand inference into multi-agent RL are well-motivated.

2. Technical soundness: The architecture components (hierarchical structure, gradient inference) are reasonable and consistent with recent developments in hierarchical and graph-based RL.

3. Empirical promise: The reported results show meaningful improvements in the studied domain, suggesting that the proposed components contribute to performance gains.

4. Readable exposition: The paper is generally well written and structured, which helps readers follow the ideas and rationale behind HGRL.

**Weaknesses:**

1. Overstated generality: The paper claims to propose a general framework for dynamic resource allocation, but all problem formulations, environments, and experiments are grid-based and specialized to a ride-hailing scenario. The contribution should be framed more narrowly or supported with experiments from additional domains to substantiate general applicability.

2. Limited experimental breadth: All numerical evidence focuses on ride-hailing. To claim domain generality, evaluation on other DRAP settings (e.g., energy systems, logistics networks) would be essential. Moreover, the baselines appear generic (e.g., PPO, DQN), while stronger domain-specific RL methods for ride-hailing exist and should be included for fair benchmarking.

3. Lack of comparison to optimization-based hierarchical controllers: The authors introduce a high-level agent but do not clarify why this layer should be learned rather than optimized using model-based or centralized methods. A discussion or experiment comparing the hierarchical agent to an optimization-based planner would clarify the added value of the proposed approach.

4. Partial theoretical presentation: Some formal parts are not cleanly structured — for example, a proof is given without a clearly stated theorem. Since the argument is short, it could be rewritten less formally, or alternatively, the authors should properly define the statement and its assumptions.

5. Scope of novelty: While the proposed combination of hierarchical multi-agent RL and gradient-informed inference is interesting, it builds on established paradigms (hierarchical RL, demand prediction networks). The authors should better articulate what conceptual advance distinguishes HGRL from previous hierarchical or information-sharing frameworks.

**Questions:**

1. Can you clarify what aspects of DRAPs beyond ride-hailing your method can handle without retraining or architectural modification?

2. How would HGRL perform in non-grid settings (e.g., continuous or network-based environments)?

3. What are the precise differences between GDI-Net and standard demand prediction networks used in prior multi-agent RL approaches?

4. Could a centralized optimization-based controller at the higher level achieve similar performance? If not, what prevents that?

5. Please restate the theorem you prove more clearly and specify its assumptions.

---

### Official Review · Reviewer_kiv5 · 2025-10-30

**Soundness:** 2
**Presentation:** 2
**Contribution:** 2
**Rating:** 2
**Confidence:** 4

**Summary:**

The paper studies a vehicle routing problem, applying a hierarchical reinforcement learning technique. This topic has been heavily studied in recent years. The proposed solution relies on a number of unrealistic assumptions and thus has limited applicability. The experimental results adopt the same assumptions and therefore offer only a limited contribution.

**Strengths:**

The problem is a well-studied topic, and dynamicity is an important aspect that has not yet been fully addressed.

**Weaknesses:**

1. The abstract and introduction discuss a very general resource allocation problem, whereas the problem formulation is specific to a vehicle routing problem. This mismatch should be addressed.

2. The system model assumes no congestion or shared bottlenecks in the network, so agents can operate independently. This assumption is essential for the reward formulation in Section 4.1 and the hierarchical design proposed in Section 4.3, but it is not realistic.

For example, in the TaxiBJ setting, taxis (i.e., agents) traverse roads that can be congested. An agent’s decision about which cell to select depends not only on the agent’s location and the location of demand (e.g., the passenger), but also on the state of the roads connecting the two. Moreover, an agent’s action (e.g., choosing road x to reach demand y) can change the environment (e.g., create additional congestion in some areas), which in turn affects other agents’ decisions. These interdependencies among agents’ decisions are missing from the framework.

I therefore suggest that the authors either revise the claims in the abstract and introduction to state explicitly the specific problem studied (rather than a general resource allocation problem in a dynamic environment), or extend the study to incorporate these environmental aspects.

**Questions:**

1. How does this study relate to the classical vehicle routing problem (VRP) literature?

2. How can environmental aspects (e.g., road congestion) be incorporated into the study? At a minimum, is it possible to include these factors in the experiments and observe their impact on system performance?

3. Shouldn’t the reward function also include latency? I understand that movement penalties indirectly penalize high-latency decisions, but if we consider latencies due to road congestion, that term would not be sufficient.

---

### Official Review · Reviewer_9K3i · 2025-11-02

**Soundness:** 2
**Presentation:** 2
**Contribution:** 2
**Rating:** 2
**Confidence:** 5

**Summary:**

This paper proposes a Hierarchical Gradient-Informed Reinforcement Learning (HGI-RL) framework to tackle dynamic resource allocation (DRA) problems under partial observability. The authors introduce a hierarchical architecture where global and local controllers are coordinated through gradient-informed signal passing, enabling multi-level agents to align objectives without explicit supervision.

**Strengths:**

1. The integration of gradient feedback across hierarchical levels is a strong conceptual contribution. It enhances coordination between global and local policies and mitigates policy interference—a long-standing challenge in hierarchical RL.
2. The hierarchical decomposition allows the algorithm to scale efficiently to large and heterogeneous multi-agent systems. The global–local separation reduces coupling and improves learning stability across distributed agents.

**Weaknesses:**

1. All experiments are conducted in simulated environments. The paper does not include deployment or testbed studies to demonstrate robustness under real-world uncertainties such as communication delays, stochastic dynamics, or gradient noise.
2. The evaluation omits several recent and relevant baselines, including meta-RL approaches (e.g., PEARL, MAML, Meta-Hierarchical Reinforcement Learning (MHRL)), graph-based resource allocators (GNN-RL), and federated optimization frameworks. Including these would provide a stronger benchmark for the claimed state-of-the-art performance.
3. The hierarchical gradient-sharing mechanism increases both computational and communication complexity. However, the paper does not provide quantitative analysis of runtime, sample efficiency, or scaling behavior with system size.
4. The framework assumes a regular grid structure for spatial representation. This assumption rarely holds in real-world DRA settings, where topology and resource connectivity are irregular. The applicability of the approach to irregular structures is not discussed.
5. While the intuition behind hierarchical gradient propagation is well-motivated, the paper lacks formal convergence guarantees, complexity analysis, or optimality bounds. This limits theoretical confidence in the method’s generality and stability.

**Questions:**

1. The paper states that GDI-Net generates demand maps at multiple spatial resolutions via resolution-specific pooling. However, Equations (3)–(7) appear to assume a fixed resolution. How does the method ensure consistency across resolutions? Could this cause misalignment between hierarchical levels?
2. In Equation (1), $ D_{t,g} $ should depend on $n$. Otherwise, its formulation is incomplete or ambiguous. Please clarify this dependency.
3. How is the number of hierarchical levels $L$ determined? Is it fixed or adaptively chosen based on system complexity or observation scale? How does it affect model performance? Please include an ablation study to quantify its impact.
4. What does $C$ represent in Equation (4)? Its definition is unclear.
5. How is the parameter w×w defined, and how does it affect model performance? Please include an ablation study to quantify its impact.
6. Similarly, how is a×a defined in Equation (4)? Its functional role and performance influence should be analyzed through ablation.
The reward term in Equation (7) appears to depend solely on distance rather than the cost function $C$. Could the authors clarify this design? The formulation seems inconsistent with the cost-driven optimization objective.
7. How does HGI-RL scale in real-world large-scale systems (e.g., multi-cell wireless scheduling, cloud resource management)? Have you measured latency per decision step?
8. How are gradients exchanged between hierarchical levels implemented in practice, synchronously, asynchronously, or via averaging? How does this choice affect stability and convergence?
9. Why were recent meta-RL and GNN-based approaches omitted from evaluation? Could inclusion of these baselines significantly alter the reported performance gap?

**Details Of Ethics Concerns:**

No.

---

### Official Review · Reviewer_RyD6 · 2025-11-03

**Soundness:** 2
**Presentation:** 2
**Contribution:** 1
**Rating:** 2
**Confidence:** 5

**Summary:**

The work addresses the problem of dynamic resource allocation in a multiagent and partially observable setting. To address the challenge the issue of scalability when extending to multiagent resource allocation settings, the work decomposes into multiscale problems, using hierarchical RL (an extension of value based decomposition method to the DRAP problem) to solve the problem.

**Strengths:**

The problem of resource allocation is practical and has been addressed in several past works.

**Weaknesses:**

The problem of resource allocation is practical and has been addressed in several past works.

The issue of scalability and partial observability have been addressed before, the whole sub-area of mean field RL and collective MARL is used to address it. I find a proper comparisons to such prior work is missing:

Credit Assignment For Collective Multiagent RL With Global Rewards. Neurips 2018.
Collective Multiagent Sequential Decision Making Under Uncertainty. AAAI 2017.

There are several papers for the taxi fleet optimization, which are not cited and compared to:
Online spatio-temporal matching in stochastic and dynamic domains. AIJ 2018

Several such works address the challenge of scalability and partial observability using count based methods.

The problem statement in section 3 needs more convincing as to why this is a general problem framework that can address a wide range of resource allocation problems. Currently, the problem statement is too simplistic and ignores major real world settings such as constraints (resources can be limited, demands need to be served within a specific time, fairness of resource allocation  etc). Currently the idealized setting presented in section 3 is not general enough to capture diversity of real world resource allocation problems. As a result, it it unclear why addressing this formulation is impactful.

Another issue is unclear notational issues in section 3 and unclear assumptions. What is w, what is “spatial service window”? Are resources renewable/non-renewable, can the method work for both types of resources? why a grid environment is used, why can’t a general map be used? Similarly, is discretization necessary, and how to do it in a real world setting? In reality, for taxi fleet optimization, roads are a continuous segment. How large should be such grid-based discretization?

Dynamic resource allocation is a problem that has been widely addressed in the OR and stochastic optimization and planing  literature. However, I did not see a discussion and comparison with the formulation used in stochastic optimization literature. A proper review justifying why the formulation in section 3 is a general and impactful formulation is required.

In terms of strength of the contribution, most of the work presents modeling of DRAP, defining reward function, hierarchy. The MARL method is a straightforward extension of the additive value decomposition networks. The work would read better by added novelty that can address limitations of VDNs in this setting.

Empirically, the evaluation of the work is somewhat weak. Only 7 days of data is used, and other examples are toy examples. There are several large taxi datasets available such as NYC taxi trip data. It would be more significant if the work addresses a realistic, large taxi dataset.

**Questions:**

See previous section.

---

### Note · Authors · 2025-12-07

**Comment:**

We thank all reviewers and the area chair for their time, thoughtful assessments, and constructive suggestions on our submission #16778. We deeply appreciate the effort invested in reading our work and providing detailed feedback.

However, after careful consideration and internal discussion, we have decided to withdraw the current version of the submission. Several key critiques appear to stem not from technical shortcomings, but from mismatches between the reviewers’ interpretations and our intended framing—particularly concerning baseline coverage and the scope of the experimental assumptions.

We would like to offer two clarifications again, which seem to have been misunderstood:

**1. Clarification on baselines and comparison methods**
We thank Reviewers RyD6 and 9K3i for highlighting the importance of including methods such as collective MARL, meta-RL, and GNN-based approaches as comparisons. These suggestions are valuable. We would also like to note that the submitted version already includes baselines aligned with these directions:

- For **collective / credit-assignment based MARL**, we included **Hierarchical Mean-Field (HMF) **[1] (2023), which is a more recent representative compared with older works such as “Credit Assignment for Collective Multi-Agent RL with Global Rewards” (NeurIPS 2018).
- For **meta-RL and GNN-based resource allocation**, we included **MDPO**[2] (2024), which incorporates meta-learning and GNN-based coordination, and is a recent and powerful approach under different scenarios.

These baselines were documented in Appendix D, due to the page limit.

References:
[1] Yu, C. (2023). Hierarchical mean-field deep reinforcement learning for large-scale multiagent systems. AAAI Conference on Artificial Intelligence, 37(10), 11744–11752.

[2] Ma, C., Li, A., Du, Y., Dong, H., & Yang, Y. (2024). Efficient and scalable reinforcement learning for large-scale network control. Nature Machine Intelligence, 6(9), 1006–1020.

**2. Clarification on experimental assumptions and domain scope**
We thank Reviewer kiv5 for the insightful comments regarding congestion modeling and interaction coupling. These factors are indeed crucial for taxi routing or ride-hailing environments. However, our work focuses on **dynamic allocation of mobile service facilities** (e.g., mobile charging stations, emergency units), rather than taxis or ride-hailing fleets themselves. Such agents do not induce congestion in the same manner, and their decision-making topology differs significantly from VRP or ride-hailing setups.


We again thank the reviewers and the AC for their constructive feedback.

**Withdrawal Confirmation:**

I have read and agree with the venue's withdrawal policy on behalf of myself and my co-authors.